# Co-Expression of Podoplanin and CD44 in Proliferative Vitreoretinopathy Epiretinal Membranes

**DOI:** 10.3390/ijms24119728

**Published:** 2023-06-04

**Authors:** Denise Bonente, Laura Bianchi, Rossana De Salvo, Claudio Nicoletti, Elena De Benedetto, Tommaso Bacci, Luca Bini, Giovanni Inzalaco, Lorenzo Franci, Mario Chiariello, Gian Marco Tosi, Eugenio Bertelli, Virginia Barone

**Affiliations:** 1Department of Life Sciences, University of Siena, Via A. Moro 2, 53100 Siena, Italy; denise.bonente@student.unisi.it; 2Department of Molecular and Developmental Medicine, University of Siena, Via A. Moro 2, 53100 Siena, Italy; claudio.nicoletti@unisi.it (C.N.); eugenio.bertelli@unisi.it (E.B.); virginia.barone@unisi.it (V.B.); 3Section of Functional Proteomics, Department of Life Sciences, University of Siena, Via A. Moro 2, 53100 Siena, Italy; rossana.desalvo@student.unisi.it (R.D.S.); luca.bini@unisi.it (L.B.); 4Department of Medicine, Surgery and Neuroscience, University of Siena, Viale Mario Bracci 16, 53100 Siena, Italyosammot.iccab@gmail.com (T.B.); gianmarco.tosi@unisi.it (G.M.T.); 5Core Research Laboratory (CRL), Istituto per lo Studio, la Prevenzione e la Rete Oncologica (ISPRO), Via Fiorentina 1, 53100 Siena, Italy; giov.inzalaco@gmail.com (G.I.); lorenzo.franci@ifc.cnr.it (L.F.); mario.chiariello@cnr.it (M.C.); 6Istituto di Fisiologia Clinica (IFC), Consiglio Nazionale delle Ricerche (CNR), Via Fiorentina 1, 53100 Siena, Italy; 7Department of Medical Biotechnologies, University of Siena, Viale Mario Bracci 16, 53100 Siena, Italy

**Keywords:** podoplanin (PDPN), cluster of differentiation 44 (CD44), epiretinal membrane (ERM), proliferative vitreoretinopathy, interactomics, transdifferentiation, fibrosis, cell migration

## Abstract

Epiretinal membranes (ERMs) are sheets of tissue that pathologically develop in the vitreoretinal interface leading to progressive vision loss. They are formed by different cell types and by an exuberant deposition of extracellular matrix proteins. Recently, we reviewed ERMs’ extracellular matrix components to better understand molecular dysfunctions that trigger and fuel the onset and development of this disease. The bioinformatics approach we applied delineated a comprehensive overview on this fibrocellular tissue and on critical proteins that could really impact ERM physiopathology. Our interactomic analysis proposed the hyaluronic-acid-receptor cluster of differentiation 44 (CD44) as a central regulator of ERM aberrant dynamics and progression. Interestingly, the interaction between CD44 and podoplanin (PDPN) was shown to promote directional migration in epithelial cells. PDPN is a glycoprotein overexpressed in various cancers and a growing body of evidence indicates its relevant function in several fibrotic and inflammatory pathologies. The binding of PDPN to partner proteins and/or its ligand results in the modulation of signaling pathways regulating proliferation, contractility, migration, epithelial–mesenchymal transition, and extracellular matrix remodeling, all processes that are vital in ERM formation. In this context, the understanding of the PDPN role can help to modulate signaling during fibrosis, hence opening a new line of therapy.

## 1. Introduction

Epiretinal membranes (ERMs) are pathologic sheets of fibrocellular tissue that develop at the vitreoretinal interface [1]. When they are located in front of the fovea and show a retracting behavior, ERMs generate a clinical condition known as macular pucker, which is characterized by a profound disruption of the retinal architecture and by the generation of retinal foldings. The clinical relevance of ERMs is variable as they can be either asymptomatic or sight-threatening [2,3]. Nevertheless, ERMs are quite common, with a prevalence ranging between 2.2% and 28.9% depending on the statistical surveys [2,4,5]. A projection based on the Beaver Dam Eye Study estimates that about 30 million people in the US are likely affected by ERMs at least in one eye and that 6.8 million of them should show signs of retinal folding [6].

ERMs can be classified according to a gradient of increasing clinical severity as cellophane macular reflex or preretinal macular fibrosis [2,6]. A distinction based on the etiology, on the other hand, acknowledges idiopathic ERMs when they develop inside an otherwise healthy eye and secondary ERMs when they are generated because of systemic (i.e., diabetes) or local disorders (e.g., trauma, retinal detachment, and macular hole) [7].

The histopathology of ERMs was extensively studied. Their extracellular matrix (ECM) component includes collagens and basement membrane proteins, which were recently investigated even at the level of their chain composition [8,9]. The panel of proteins found in the ECM of ERMs is, however, far larger than the mere structural molecules. A vast network of interactions is actually created among structural, adhesive, and matricellular proteins, as well as with many locally secreted growth factors/chemokines, proteases, and inhibitors of proteases. All these molecules form the milieu of ERM ECM and cross-talk with ERM cells through a vast repertoire of receptors, ultimately influencing cell behavior [10].

Cells dwelling in ERMs are a heterogeneous population with ultrastructural features and molecular markers belonging to several different cell types, including hyalocytes, Müller cells, fibroblasts, astrocytes, and retinal pigment epithelial cells (RPE) [11,12,13]. Regardless of their lineage, ERM cells transdifferentiate toward a myofibroblastic phenotype by expressing mesenchymal markers [14,15,16,17] and becoming capable of contraction and ECM protein secretion [17,18].

During the early stages of membrane formation and progression, ERM cells are bound to migrate on and along the vitreoretinal interface. The identification of molecules positively involved in promoting cell migration in ERMs is therefore important to understand some steps in the chain of events leading to the formation of macular puckers and, in particular, to find possible targets for new therapeutic strategies, which may interfere with the disorder progression.

Podoplanin (PDPN), also known as D2-40, gp36, T1α, aggrus, and OTS-8, is a transmembrane receptor glycoprotein that is involved in several contexts of cell migration, including tumor cells [19,20], wound healing [21], and cancer associated fibroblasts [22,23]. Additionally, the cluster of differentiation 44 (CD44) may have pivotal roles in ERM formation being implied in cytoskeleton rearrangement, cell proliferation, adhesion, migration, angiogenesis, and inflammation [24]. Interestingly, the PDPN–CD44 interaction was proven to be crucial for the directional persistence of motility in epithelial cell cancer [19,20].

Here, we report that, not depending on their etiology, ERMs produce PDPN and the majority of PDPN^+^ cells also synthesize CD44, thus suggesting their possible interaction and involvement in cell migration during ERM onset and development. Taking advantage of the network we previously published in a study on ERM ECM interactome [10], we also attempted to predict the functional involvement of PDPN in ERM and we investigated the presence and immunolocalization of some of the proteins that the predictive analysis highlighted as entailed in PDPN regulation and ERM formation.

## 2. Results

### 2.1. PDPN and CD44 Detection in ERMs

Laser scanning confocal microscopy of ERM sections showed a consistent presence of PDPN in all samples, regardless of their idiopathic or secondary nature. A variable number of PDPN^+^ cells were constantly present in all membranes. Characterization of PDPN^+^ cells was not very informative, as PDPN could be found expressed in a variety of cells, namely vimentin (VIM)^−^/cytokeratin (CK)^+^, VIM^−^/CK^−^, VIM^+^/CK^+^, VIM^+^/Glial-fibrillary-acidic-protein (GFAP)^+^, VIM^+^/GFAP^−^, and VIM^−^/GFAP^−^ cells.

However, the same intermediate filament expression profile could be seen also in some PDPN^−^ cells. Even PDPN location was not a constant feature, as it could be found either cytoplasmic or on cell membrane (Figure 1a).

CD44 was previously reported in ERMs [25,26]. As it probably plays a relevant role in the development of ERMs [10], we analyzed its occurrence in our samples, trying to characterize positive cells. CD44 appeared broadly present in all tested membranes. It was observed in VIM^+^/CK^+^ cells and in VIM^+^/CK^−^ cells. Indeed, very few cells skipped CD44 expression (Figure 1b). Double labeling experiments, aiming to investigate if CD44 and PDPN are present in the same cell, revealed that the great majority of ERM cells were immunoreactive for both CD44 and PDPN. Only very few cells were either CD44^+^/PDPN^−^ or CD44^−^/PDPN^+^ (Figure 2).

RT-qPCR on 9 patients confirmed that all the analyzed ERMs contained CD44 and PDPN mRNAs, though with different expression levels (Figure 2b). In keeping with results observed by immunofluorescence, levels of expression were variable probably as a result of the different percentage of PDPN^+^ cells within each membrane.

### 2.2. Functional Prediction of PDPN Role in ERM

Previously, we performed a functional predictive study on ERM ECM and ECM-related proteins [10] by reanalysing ERM proteins identified by [26]. Because of the applied methods and parameters, a number of ERM proteins bona fide escaped the [26] analysis, as it probably happened to PDPN. Nonetheless, with the intent to evaluate the possible inclusion and the functional relevance of the latter in the direct interaction network (DIN) we generated in [10], PDPN was added to the [10]-net as described in Materials and Methods (Figure 3). It entered the network by the addition of 21 non-experimental proteins (PDPN first niche of interaction; FNI) that function as molecular bridges to correlate PDPN to other net experimental proteins (PDPN second niche of interaction; SNI) (Figure 3).

With the intent to make easier the visualization of FNI and SNI proteins, we applied the trace mode visualization that marks the selected net protein(s) and its (their) FNI factors by making all the unselected net objects semi-transparent and highlighting any interaction (edge) that the selected protein(s) establishes. The Figure 4 network enlargement displayed in Figure 4a clearly shows the 21 proteins of the PDPN FNI. Most of these added nodes are transcription factors (TFs) and the majority of them link the high mobility group protein B1 (HMGB1) to PDPN (Figure 4b). In addition, among the MetaCore added TFs, Krueppel-like factor 4 (KLF4), transcription factor E2F1 (E2F1), protein cFos (cFos), and the signal transducer and activator of transcription 3 (STAT3) are positively controlled by the epidermal growth factor receptor (EGFR), and they functionally link the latter to PDPN, with cFos and STAT3 having an inductive effect on PDPN (Figure 4c). This is of particular relevance since EGFR resulted in one of the most important proteins (central hubs) in the [10]-net. Interestingly, CD44 positively controls EGFR as well as STAT3. CD44 further correlates with the PDPN FNI by inducing the mothers against decapentaplegic homolog 2 (SMAD2), homeobox protein NANOG (NANOG), and transcription factor SOX-2 (SOX2), with SOX2 positively controlling PDPN expression (Figure 4d) [27]. Worthy of note, STAT3 also correlates PDPN with vascular endothelial growth factor receptor 1 (VEGFR-1), another relevant [10]-net hub in delineating the biomolecular processes active in ERM formation and development.

Because of its involvement in tissue fibrosis, and even more importantly, in proliferative vitreoretinopathy development (PVR) [28], we also evaluated the functional involvement of the transcriptional regulator yes associated protein 1 (YAP1) in the PDPN/[10]-network. Similarly to PDPN, YAP1 was not identified by [26]. Nonetheless, when we added it to the PDPN/[10]-network (Figure 3), YAP1 established direct interactions with the original net node HMGB1 (Figure 5a). In addition, it functionally correlates to PDPN through 13 proteins from the PDPN FNI (Figure 5b). Among them, SOX2 and NANOG link YAP1 to CD44, while E2F1, cFos, KLF4, and β-catenin connect YAP1 to EGFR (Figure 5c). Finally, STAT3 mediates YAP1 cross-linking to both CD44 and EGFR (Figure 5d).

### 2.3. Expression and Immunolocalization of TFs Regulating PDPN

Based on the FNI of PDPN, we focused on TFs linking the HMGB1 to PDPN. First, we investigated HMGB1 expression in ERM samples by RT-qPCR. Results show that HMGB1 was always expressed with little differences among patients (Figure 6a). Subsequently, we selected, among the non-experimental hubs added by the software to include PDPN into the [10]-net, the TFs SOX2 and STAT3 in reason of their functional relevance in the YAP/PDPN/[10]-net and for their known involvement in fibrotic processes. Finally, we also investigated the YAP1 presence and localization in ERM. Immunofluorescence experiments revealed that, although they were widely present in ERM cells, SOX2, STAT3, and YAP1 were mainly located in the cytoplasm. However, a limited immunoreactive signal could be detected even in the nuclei (Figure 6b).

## 3. Discussion

PDPN is a transmembrane protein that is expressed mainly, but not exclusively, in lymphatic endothelial cells, type-I pneumocytes, and podocytes [29,30,31]. More frequently, PDPN is considered a marker of lymphatic endothelial cells that is required for correct lung and lymphatics development [31,32,33]. Its role remains more elusive in adult tissues, where it could act as a regulator of cell morphology. For instance, nephrosis is characterized by loss of podocyte regular morphology with foot process effacement. In experimental models of nephrosis, such as puromycin aminonucleoside-induced nephrosis and in Dahl SS rats [30,34], this striking alteration is preceded by PDPN down-regulation. As foot process effacement is a RhoA-dependent event [35] and PDPN displays an inhibitory activity on RhoA, it was suggested that one role for PDPN could be the promotion of foot process formation [34].

Depending on the cell type, PDPN binding to partner proteins also attunes numerous cellular events, including proliferation, contractility, epithelial-mesenchymal transition (EMT), and remodeling of the ECM [20]. However, an acknowledged major role for PDPN is to mediate cell migration in several contexts, including keratinocyte, cancer-associated fibroblast, mesenchymal stem cell, and myocardial infarction-activated fibroblast migration [23,36,37,38]. In general, PDPN^+^ cells show a more pronounced predisposition to migrate. This is particularly important when PDPN is expressed by cancer cells, as it is actually associated with more aggressive tumor phenotypes [20].

Here, we proved that PDPN is expressed by all the ERMs we analyzed, regardless of their etiology. This finding is of particular relevance, as it is still unclear which type of signals and molecules induces cells to move to the vitreoretinal interface and to generate ERMs. In light of PDPN presence in ERMs and according to the above statements, we attempted a predictive functional characterization of PDPN in ERMs by its inclusion into a network we previously built [10] by processing ECM and ECM-related proteins, selected by a mass spectrometry-based study on ERMs [26]. PDPN was significantly integrated into the net by the addition of TFs highly relevant in fibrosis, trans-differentiation, and cell migration processes, e.g., KLF4, SOX2, NANOG, STAT3, and OCT3/4. Moreover, through a number of its FNI proteins, PDPN also indirectly interacts with factors that we highlighted as critically active in ERM pathophysiology [10]. In particular, EGFR, CD44, and VEGFR-1 correlate with PDPN via STAT3. The PDPN cross-talk with EGFR further stresses its possible involvement in ERM forming cell proliferation and migration [39,40], as well as in fibrosis [41,42]. PDPN could also promote ERM cell migration in conjunction with its molecular partner CD44, which we also detected in all the analyzed samples. CD44 is a molecule implied in cytoskeleton rearrangement, cell proliferation, adhesion, migration, angiogenesis, and inflammation [24]. Its role is well known in tumor cell migration. However, CD44 is also implicated in fibroblast and endothelial cell migration [43,44]. Transfection experiments showed that it can directly bind PDPN [45] and that CD44/PDPN interaction is crucial for the directional persistence of motility in epithelial cancer cells [19,20]. Other experiments, e.g., immunoprecipitation, would be useful to prove direct interaction between endogenous PDPN and CD44. Nonetheless, the small amounts of proteins recovered from ERMs define consistent restrictions for direct assays on protein–protein interaction.

As already suggested, PDPN may contribute to ERM formation by promoting EMT. De-differentiation of ERM cells towards a myofibroblastic phenotype is a hallmark of all ERMs [14,15,16,17], and it is worth noting that the ectopic expression of PDPN in epithelial cells promotes their switch from an epithelial to a fibroblast-like phenotype [46]. Interestingly, TGF-β, a key factor in EMT [47] and in ERM pathophysiology [18,48], is capable of inducing PDPN expression [49]. Furthermore, the above-mentioned PDPN FNI proteins SOX2, NANOG, OCT3/4, and KLF4 are crucial factors also in inducing pluripotency in somatic cells [50]. Consequently, PDPN occurrence in ERMs could represent just one of the many events that outline trans-differentiation and pro-fibrotic processes underlying ERM formation.

Therefore, we tested, by immunofluorescence, the occurrence of SOX2 and STAT3, two key TFs supposed to modulate PDPN expression. The main presence of these TFs in the ERM cell cytoplasm suggests them as active in different roles, depending on different cell state and/or stimuli, during retinal fibrosis. In fact, context-specific cytoplasmic roles were described for STAT3, independently of its nuclear functions, widely related to fibrosis [51,52,53,54]. In addition to its involvement in cellular redox state control, senescence and apoptosis, Ca^2+^ homeostasis, and energy balance under stress conditions [53], cytoplasmic STAT3 localizes at focal adhesions in migrating cells [55,56] and concurs to modulate microtubule (de)polymerization [57], as well as stress fiber synthesis in fibroblasts [58]. However, though not prominent, SOX2 and STAT3 migration into the nuclei may bear a certain importance in ERMs. Both SOX2 and STAT3, in fact, were shown to act as TFs for collagen I synthesis in several fibrogenic settings. In particular STAT3 works as a non-canonical TF in response to TGF-β induction [52,59,60].

In relation to its involvement in PVR and to its relevant inclusion in the PDPN/[10]-net, we also investigated the presence and cellular distribution of YAP1 by immunofluorescence. Similarly to SOX2 and STAT3, YAP1 was observed in the cytoplasm and to a lesser degree in the nuclei. This protein is a transcriptional coactivator involved in mechanotransduction and in the Hippo pathway and was reported modulating gene expression in response to tissue stiffness. In various fibrotic disorders, YAP1, along with tafazzin (TAZ), coordinates the injury-induced myofibroblast phenotype and transduces extracellular mechanical signals independently or by interacting with other profibrotic pathways, such as TGF-β, EGFR, and Wnt signaling [61]. It is noteworthy that in a diabetic rat model, YAP1 activation was found to induce EMT of RPE cells, and its nuclear localization correlates with TGF-β-induced retinal fibrosis by promoting the fibrogenic activity of Müller cells [28]. On the other hand, the interaction of YAP1 with cytosolic partners could represent an additional signaling pathway in the development of ERM. In agreement, YAP1 was described to sequester β-catenin in the cytosol, thus inhibiting the Wnt/β-catenin pathway The Hippo/YAP1 and Wnt/β-catenin signaling pathways also affect each other. In the absence of Wnt ligand, the two transcriptional coactivators of the Hippo pathway, YAP and TAZ, are in fact sequestered in the cytoplasm by β-catenin, glycogen synthase kinase 3 (GSK3), and the Axin1 destruction complex, which prevents them from translocating into the nucleus [62].

YAP1 entered the PDPN/[10]-net by directly interacting with the HMGB1 and through numerous protein nodes that were added by the MetaCore software to allow PDPN inclusion into the original [10]-network. HMGB1, in turn, is critical also for PDPN, as all PDPN FNI proteins are under the direct control of this ubiquitously expressed protein. HMGB1 plays distinctive roles in different subcellular compartments [63]. In the cytosol, it is involved in oxidative stress-mediated autophagy [64] and acts as a sensor and/or chaperone for immunogenic nucleic acids, which are responsible for the activation of the Toll-like receptor 9 (TLR9) immune response [65]. Furthermore, HMGB1 can be released in the extracellular environment, where it functions as a signal molecule capable of mediating several functions, including the inflammatory response in the early phases of tissue injury by the recruitment of inflammatory cells [66], cytokine and chemokine release [67], cytoskeleton (re)organization and cell migration [68], and EMT induction [69]. Interestingly, Zhang et al. [70] found that radiotherapy induces the release of HMGB1 in pancreatic cancer cells, thus promoting their dedifferentiation into the cancer stem cells phenotype via the TLR2/Hippo/YAP1 pathway.

From a clinical viewpoint, PDPN protein expression in ERMs may give us a clue for possible novel pharmacological treatments of ERM progression. PDPN, in fact, is a true receptor whose engagement by its ligand, the C-type lectin-like receptor 2 (CLEC-2) [71], could affect ERM cell behavior. CLEC-2 is found on platelets and also in plasma as a soluble form [72]. It acts on PDPN^+^ cells, inducing the dephosphorylation of ezrin, radixin, and moesin, and uncoupling PDPN from the underlying actin cytoskeleton. In cancer-unrelated contexts, interactions between platelets and cells, via CLEC-2 binding to PDPN, leads to controversial results. In wound healing and lymphatic vessel development, CLEC-2/PDPN binding reduces, respectively, keratinocyte and lymphatic endothelial cell migration [36,73]. In contrast, CLEC-2/PDPN binding was shown to induce dendritic cell migration [74]. However, in the latter scenario, moving cells are those bearing CLEC-2 rather than PDPN^+^ elements. The privileged retinal environment, protected by an inner and an outer blood–retinal barrier [75], excludes ERM cells from interacting with circulating platelets and likely also precludes soluble CLEC-2 to reach ERMs. As ERM resident cells are mostly PDPN^+^ cells that proliferate and migrate in a CLEC-2-free environment, they might be responsive to exogenously introduced CLEC-2, possibly decreasing their migratory attitude. Studies are currently in progress to test this hypothesis.

## 4. Materials and Methods

### 4.1. PDPN Functional Inclusion into ECM and ECM-Related Proteins from ERMs

Functional data processing was performed using the MetaCore v21.3 (Clarivate Analytics, Boston, MA, USA) integrated software suite for functional analysis of experimental data. The MetaCore network building tool is based on a manually annotated and periodically updated database of protein interactions and metabolic reactions in physiological and pathological states from scientific literature.

Gene names of the 141 selected proteins from [26] were submitted to MetaCore and co-processed by the “direct interaction” algorithm, which allows for functionally correlating only factors that, according to the MetaCore database, are involved in direct interactions [10]. Successively, we used the “shortest path” algorithm (SPA), set to “high trust interaction”, and allowed the addition of only one non-experimental factor to crosslink PDPN to DIN-experimental proteins. Likewise, we also inserted into the PDPN/[10]-network YAP1. The resulting networks were built limiting protein processes to individual proteins, excluding their involvement in multimeric complexes, and avoiding canonical pathways. The generated pathway maps were prioritized according to their statistical significance (*p* ≤ 0.001) and the networks were graphically visualized as nodes (proteins) and edges (relationship between proteins).

Similarly to previous studies, this analysis provides high quality and reliable functional correlations that characterize the biological contest in which the proteins of interest act [10,76,77,78,79].

### 4.2. Patients and Tissues

Twenty-three patients affected by ERM were collected. Of these, 18 were idiopathic (9 males and 9 females, average age 69.7) and 5 were secondary to retinal detachment (5 male, average age 71.8). Patients were subjected to primary 25-gauge pars plana vitrectomy by the same surgeon (G.M.T) without intraocular complications at the local hospital (AOUS Scotte, Siena, Italy). ERMs were stained according to surgeon preference and were excised using Grieshaber Revolution forceps (Alcon Laboratories Inc., Fort Worth, TX, USA). At the end of the procedure, fluid/air exchange was performed in all patients. Ethics committee approval and written informed consent were obtained for all patients.

For immunofluorescence experiments, fourteen ERMs were formalin-fixed and paraffin-embedded or OCT-frozen and acetone-fixed for sectioning. Seven μm sections were cut from each sample. Nine ERMs were collected in a RNA preservation medium in order to extract total RNA and perform a RT-qPCR.

### 4.3. Confocal Microscopy

For protein immunofluorescent localization by confocal microscopy, the following primary antibodies were used: mouse anti-podoplanin monoclonal antibody (clone D2-40) from Dako (Glostrup, Denmark); rabbit anti-CD44 and anti-SOX2 polyclonal antibodies from Sigma (St. Louis, MO, USA); goat anti-vimentin polyclonal antibody [80] kindly gifted from Prof. Peter Traub (Max-Planck-Institut fur Zellbiologie; Rosenhof, Ladenburg, Germany); polyclonal rabbit anti-GFAP from Zymed Laboratories (San Francisco, CA, USA); rabbit anti-pan cytokeratin polyclonal antibody from Bioss Antibodies (Woburn, MA, USA); and rabbit anti β-catenin monoclonal antibody (clone D10A8) was purchased from Cell Signaling Technology (Danvers, MA, USA), whereas rabbit anti-STAT3α monoclonal antibody (clone D1A5), also from Cell Signaling Technology (Danvers, MA, USA), was a kind gift from Prof. Cristina Ulivieri (Dept of Life Sciences, University of Siena, Siena, Italy). The secondary antibodies used were: donkey TRITC-conjugated anti-goat IgG from Chemicon (Temecula, CA, USA), donkey FITC-conjugated F(ab’)2 fragment anti-mouse IgG, and donkey Cy5-conjugated F(ab’)2 fragment anti-rabbit IgG and donkey TRITC-conjugated anti-rabbit IgG from Chemicon (Temecula, CA, USA).

Immunoreactions on paraffin-embedded sections were performed upon paraffin removal. Heat-induced antigen retrieval was achieved in a heater at 98 °C in 50 mM Trizma-base (pH 9.0) for 20 min. Acetone-fixed frozen sections were directly processed for immunolabeling. The appropriate secondary antibodies conjugated with FITC, TRITC, or Cy5 were used to unveil the positive immunoreactions with primary antibodies. Cell nuclei were stained with DAPI at the end of all reactions. Controls were carried out with the same procedures, but omitting the primary antibody. Images were acquired with an LSM510 Zeiss (Jena, Germany) confocal microscope with selective multitracking excitation.

### 4.4. Quantitative Real-Time PCR (RT-qPCR)

After surgery, nine ERM samples were immediately collected in a RNA preservation medium (NucleoProtect RNA, Macherey-Nagel, Düren, Germany). Total RNA was extracted from each ERM with the NucleoSpin RNA XS kit (Macherey-Nagel, Düren, Germany) according to the manufacturer’s instructions. Then, reverse transcription into cDNA was carried out using the QuantiTect Reverse Transcription Kit (Qiagen, Hilden, Germany) in a total volume of 10 μL. The qPCR reactions were conducted using Luna Universal qPCR SYBR mastermix (New England Biolabs, Ipswich, MA, USA) on the QuantStudio 5 Real-Time PCR System (Applied Biosystems, Waltham, MA, USA). Expression levels of the selected genes were normalized to HPRT expression. Values are expressed as fold change with respect to one arbitrarily chosen ERM specimen that was used as a reference sample for all the reactions. The following primer sequences were used: human HPRT forward TGACACTGGCAAAACAATGCA and reverse GGTCCTTTTCACCAGCAAGCT; human PDPN forward GTGCCGAAGATGATGTGGTGAC and reverse GGACTGTGCTTTCTGAAGTTGGC; human CD44 forward CCCAGACGAAGACAGTCCCTGGAT and reverse CACTGGGGTGGAATGTGTCTTGGT; human HMGB1 forward GCGAAGAAACTGGGAGAGATGTG; and reverse GCATCAGGCTTTCCTTTAGCTCG.

## 5. Conclusions

PDPN is a transmembrane protein that is involved in cell migration. Its expression in ERMs could be one of the molecular events that induce cells to move to the vitreoretinal interface. PDPN has tight functional connections with CD44 and EGFR, also expressed in ERMs, outlining a pool of receptors that may all contribute to the migratory attitude of ERM cells. PDPN expression is regulated by several TFs, including SOX2, STAT3, YAP1, and HMGB1 that we detected in all tested ERMs. PDPN could be the target of pharmacological treatments aimed to inhibit cell migration and ERM progression.

## Figures and Tables

**Figure 1 ijms-24-09728-f001:**
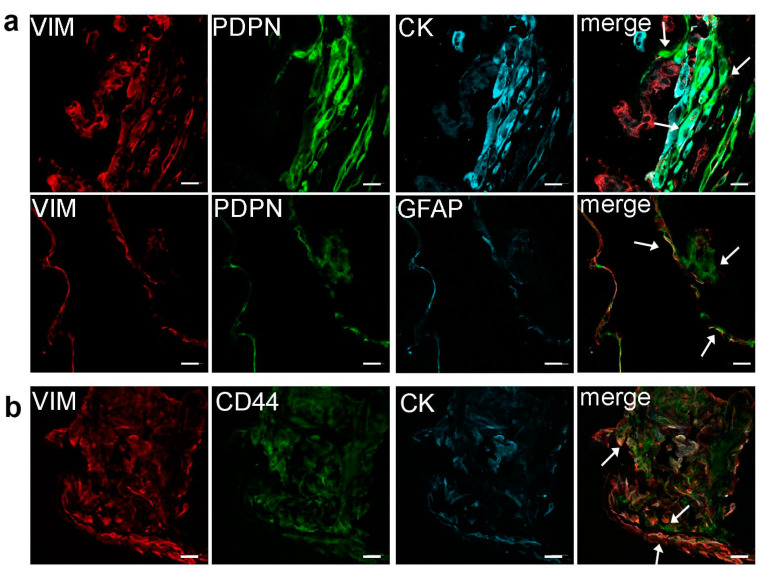
(**a**) Immunolocalization of PDPN in ERM cells showing expression of PDPN in different cells: VIM^−^/CK^−^, VIM^−^/CK^+^, VIM^+^/CK^+^ (upper row, arrows), VIM^−^/GFAP^−^, VIM^+^/GFAP^−^, and VIM^−^/GFAP^+^ (lower row, arrows). (**b**) Immunolocalization of CD44 in ERMs showing CD44 expression in both VIM^+^/CK^+^ and VIM^+^/CK^−^ cells (arrows). Scale bar 20 μm.

**Figure 2 ijms-24-09728-f002:**
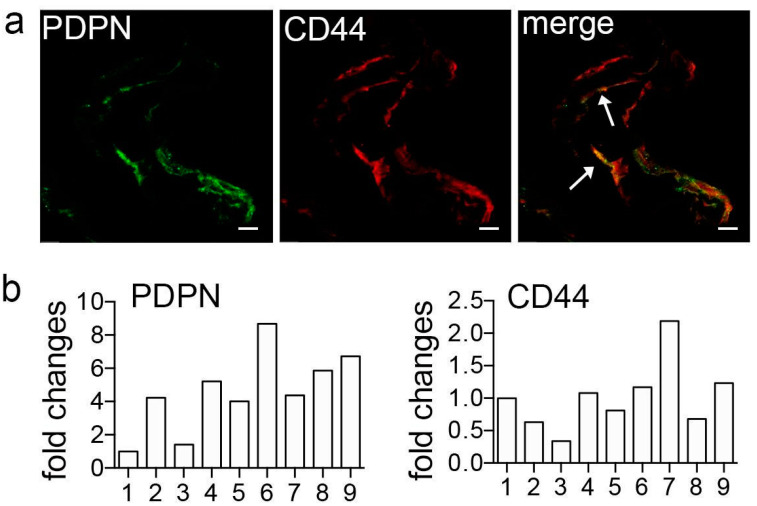
(**a**) Immunofluorescence of PDPN and CD44 showing co-expressing cells (arrows). Scale bar 20 μm. (**b**) Real-time qPCR on 9 patients (numbered from 1 to 9 on the x axis) confirmed PDPN and CD44 co-expression in all analyzed ERMs. Values are expressed as fold change with respect to one arbitrarily chosen ERM specimen that was used as reference sample for all the reactions (patient 1).

**Figure 3 ijms-24-09728-f003:**
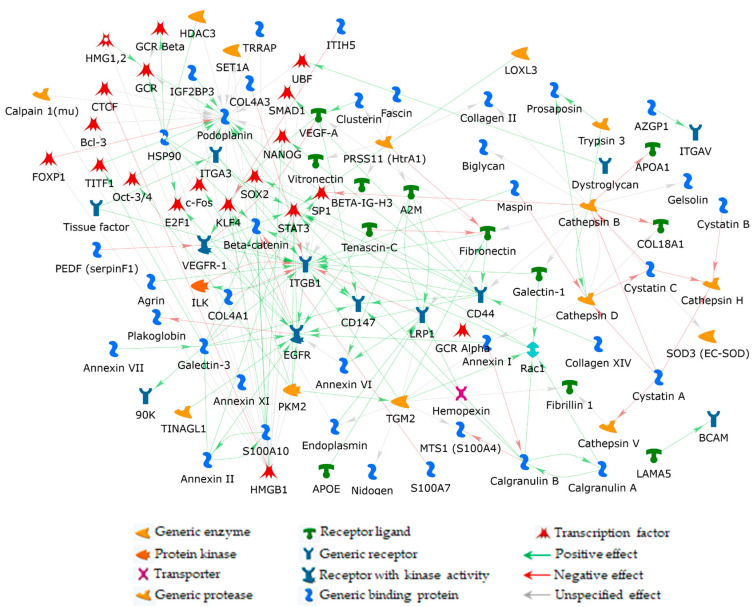
PDPN hub added to the [10]-DIN, previously obtained by processing 141 selected factors from ERM-identified proteins [26].

**Figure 4 ijms-24-09728-f004:**
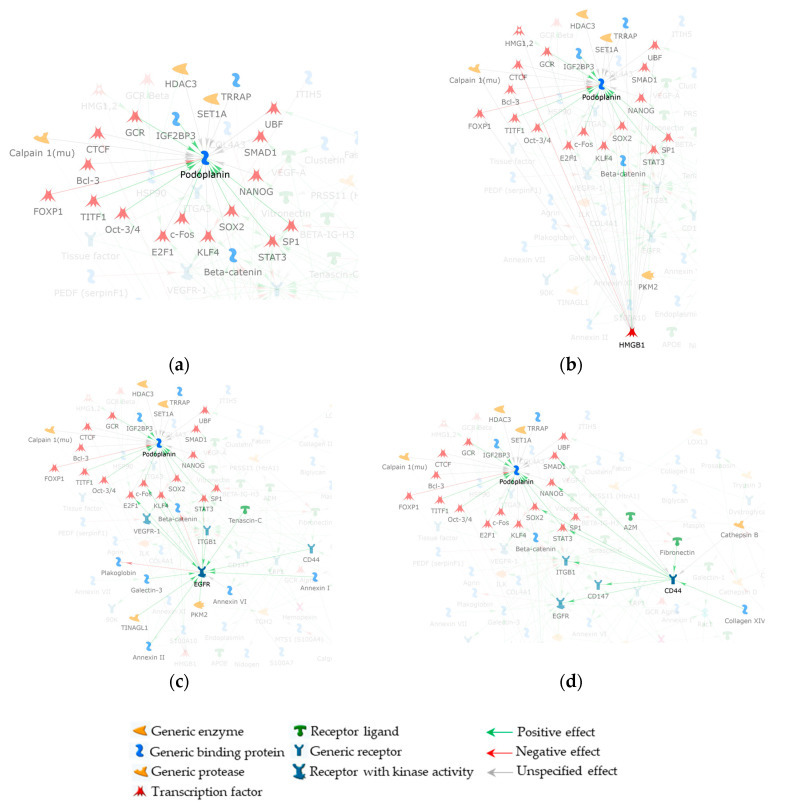
Enlargements of PDPN/[10]-network, in trace mode visualization, highlighting: (**a**) the 21 non-experimental factors added by the software to get PDPN into the net. These proteins form the FNI of PDPN and link it to at least one of the net experimental nodes; (**b**) the original net node HMGB1 that modulates the majority of the FNI proteins; (**c**) FNI proteins shared by EGFR and PDPN; and (**d**) CD44 and PDPN shared protein interactors from their FNIs. To visualize FNI proteins shared by PDPN and other net nodes, the trace mode visualization was centered on both PDPN and original net nodes that resulted as the principal interactors with the PDPN FNI (i.e., HMGB1, EGFR, and CD44). For this reason, some nodes also not belonging to the PDPN FNI (but to HMGB1, EGFR, and CD44 FNI) and not acting as a molecule bridge between PDPN and original nodes are visible in (**b**–**d**) panels.

**Figure 5 ijms-24-09728-f005:**
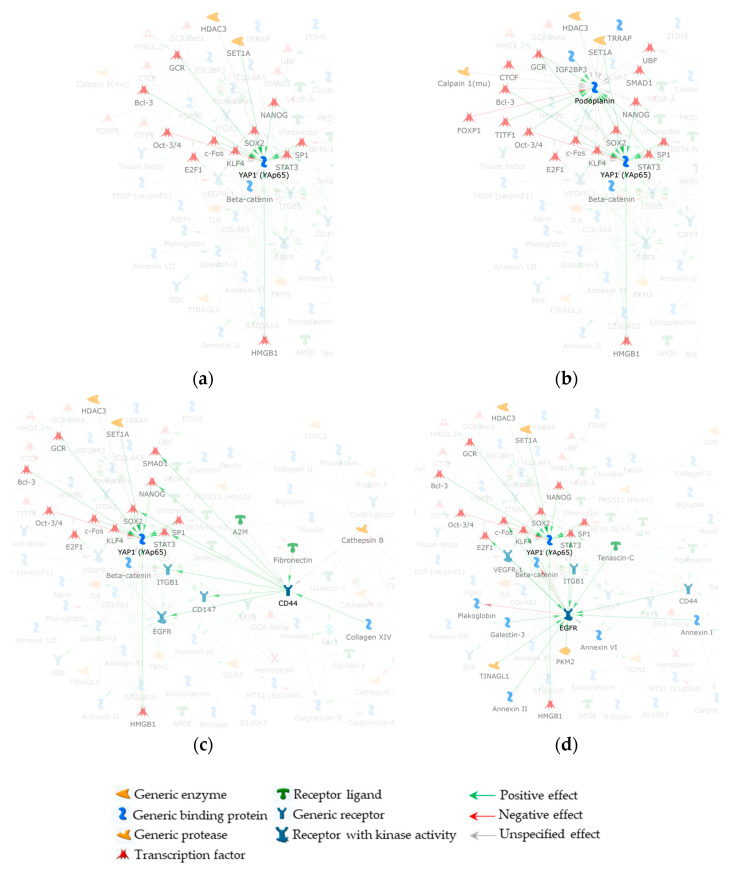
Enlargements of YAP1/PDPN/[10]-network, in trace mode visualization, highlighting: (**a**) the YAP1 FNI, including HMGB1 and 13 factors necessary for YAP1 interaction with PDPN/[10]-network nodes; (**b**) all the 13 proteins from the YAP1 FNI are shared with the PDPN FNI; (**c**) YAP1 FNI proteins shared with CD44; and (**d**) YAP1 and EGFR FNI common proteins. To visualize FNI proteins shared by YAP1 and other net nodes, the trace mode visualization was centered on both YAP1 and PDPN/[10]-network nodes that resulted as the principal interactors with the YAP1 FNI (i.e., PDPN, CD44, and EGFR). For this reason, some nodes also not belonging to the YAP1 FNI (but to PDPN, CD44, and EGFR FNI) and not acting as a molecule bridge between YAP1 and original nodes are visible in (**b**–**d**) panels.

**Figure 6 ijms-24-09728-f006:**
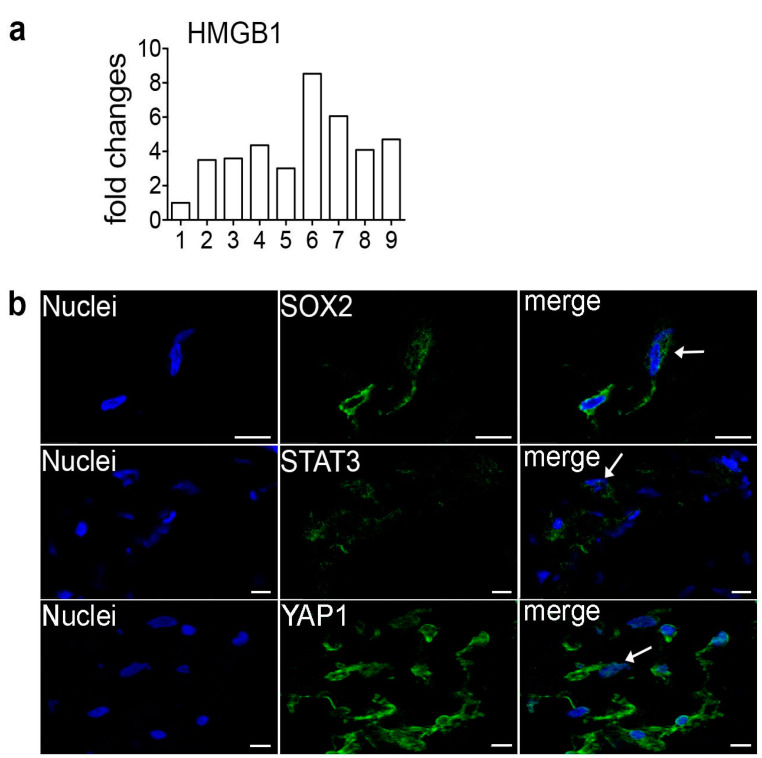
(**a**) Real-time qPCR on nine patients (x axis) revealed HMGB1 expression in all ERMs analyzed. Values are expressed as fold changes with respect to one arbitrarily chosen ERM specimen that was used as a reference sample for all the reactions (patient 1). (**b**) Immunofluorescence on ERMs with SOX2, STAT3, and YAP1 antibodies showing cytoplasmic and faint nuclear staining (arrows). Scale bar 10 μm.

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
