# Peer review of "Co-Expression of Podoplanin and CD44 in Proliferative Vitreoretinopathy Epiretinal Membranes"

_ijms, 2023, doi:10.3390/ijms24119728_

Round 1

Reviewer 1 Report

The work of Bonente et al. aims to describe the expression of PDPN and CD44 in ERMs. The title is misleading since it suggests that PDPN is a biochemically validated direct interaction partner of CD44. The manuscript is well written, and the data support the primary aim of the paper. However, the paper is full of speculations and overstatements that are born by semi-accurate in silico based interaction tables. 

The interpretation of the interaction of PDPN and CD44 is based on studying over-expressed, tagged-constructs. This gives a good idea that those proteins may interact but it is not a prove. Here, experiments that show the interaction of endogenous proteins are needed. 

Similarly, the observation that the three tx locates to the cytoplasm is just half the truth. They will be also in the nucleus most likely, as they are in all papers the authors cite. Only a proper biochemical fractionation would should whether they located in the cytoplasm only. Moreover, it would be necessary to monitor the phosphorylation status of STAT3 and YAP1.

Thus, I would prefer a more careful interpretation and discussion of a little bit perfunctory presented experimental data.

no comments

Author Response

We are submitting the revised version of the manuscript ijms-2411706 by Bonente et al. We are grateful to Reviewers for their comments as the suggested changes have improved the clarity of the manuscript.

A detailed point-by-point replay is reported below for each Reviewer #1 comment. All changes in the manuscript have been marked up by using the “Track Changes” function.

The work of Bonente et al. aims to describe the expression of PDPN and CD44 in ERMs. The title is misleading since it suggests that PDPN is a biochemically validated direct interaction partner of CD44. The manuscript is well written, and the data support the primary aim of the paper. However, the paper is full of speculations and overstatements that are born by semi-accurate in silico based interaction tables. 

We understand the Reviewer's criticism regarding the manuscript title and we changed it to “Co-expression of podoplanin and CD44 in proliferative vitreoretinopathy epiretinal membranes”.

On the other hand, we are very sorry that Rev. #1 considers the paper full of speculation. Functional bioinformatic analyses are predictive, but this does not imply that they do not reflect a real biological situation or one that is very close to reality. Based on this concept and as common practice in predictive functional analyses, we have drowned our conclusion and discussed, according to relevant literature, the obtained highly (statistically) significant networks and the main relevant nodes they highlighted. Furthermore, digressions on non-experimentally identified proteins added by the software to cross-link experimental factors, e.g. STAT3, SOX2, NANOG, E2F1, and KLF4, are of relevance and are classically present in network discussions. In fact, these proteins often corroborate the functional meaning of experimental factors by correlating them with known pathways or by shedding new lights on processes not fully understood.

The interpretation of the interaction of PDPN and CD44 is based on studying over-expressed, tagged-constructs. This gives a good idea that those proteins may interact but it is not a prove. Here, experiments that show the interaction of endogenous proteins are needed. 

We thank Rev. #1 for this comment and we fully agree with her/him. In order to avoid misunderstandings about the message we want to provide in relation to the analyses we carried out, we have modified the paper title (as reported above) and the two following sentences:

i) in the introduction section, “Here we report that, not depending on their etiology, ERMs produce PDPN and the majority of PDPN+ cells also synthesize CD44, thus suggesting a possible role of PDPN-CD44 complex in cells migration during ERM onset and development” into “Here we report that, not depending on their etiology, ERMs produce PDPN and the majority of PDPN+ cells also synthesize CD44, thus suggesting their possible interaction and involvement in cell migration during ERM onset and development” (current lines: 73-76);

ii) in the discussion section, “It directly binds PDPN [45] and CD44/PDPN interaction is crucial for directional persistence of motility in epithelial cancer cells” into “Transfection experiments showed that it can directly bind PDPN [45] and CD44/PDPN interaction is crucial for directional persistence of motility in epithelial cancer cells” (current lines: 260-262).

Still in the discussion, we also added the following sentence: “Other experiments, e.g. immunoprecipitation, would be useful to prove direct interaction between endogenous PDPN and CD44. Nonetheless, direct assays on their interaction could not be carried out since the amount of proteins retrievable from ERMs is not enough” (current lines: 262-265).

Similarly, the observation that the three tx locates to the cytoplasm is just half the truth. They will be also in the nucleus most likely, as they are in all papers the authors cite. Only a proper biochemical fractionation would should whether they located in the cytoplasm only. Moreover, it would be necessary to monitor the phosphorylation status of STAT3 and YAP1.

Thus, I would prefer a more careful interpretation and discussion of a little bit perfunctory presented experimental data.

Once again we absolutely agree with Rev. #1 and we thank her/him for this criticism. In the previous version of the manuscript we based our results on immunoreactions on paraffin embedded sections. Prompted by Reviewer’s criticism, we extended our experiments also on cryosections, previously employed only for the detection of CD44 (the antibody apparently does not work with formalin fixed paraffin-embedded sections).  In cryosections we observed that, probably gaining something in reaction sensitivity, a faint signal of the tested TFs could be observed even within the nuclei.

As a consequence, we modified Figure 6, and related legend, as well as the following sentences:

i) in the results’ section, “Immunofluorescence experiments revealed that, although they are widely present in ERM cells, SOX2, STAT3, and YAP1 remained confined to the cytoplasm as no signal was detected within the nuclei” into “Immunofluorescence experiments revealed that, although they were widely present in ERM cells, SOX2, STAT3, and YAP1 were mainly located in the cytoplasm. However, a limited immunoreactive signal could be detected even in the nuclei” (current lines: 209-212);

ii) in the discussion section, “The confined presence of these TFs in the ERM-cell cytoplasm suggests them as active in different roles, depending on different cell state and/or stimuli, during retinal fibrosis” into “The main presence of these TFs in the ERM-cell cytoplasm suggests them as active in different roles, depending on different cell state and/or stimuli, during retinal fibrosis” (current lines: 277-279);

iii) in the discussion section, “Similarly to SOX2 and STAT3, YAP1 was exclusively observed in the cytoplasm” into “Similarly to SOX2 and STAT3, YAP1 was observed in the cytoplasm and to a lesser degree in the nuclei” (current lines: 292-293).

We also added the following sentence: “However, though not prominent, SOX2 and STAT3 migration into the nuclei may bear a certain importance in ERMs. Both SOX2 and STAT3, in fact, have been shown to act as TFs for collagen I synthesis in several fibrogenic settings. In particular, STAT3 works as a non-canonical TF in response to TGF-beta induction [52,59,60]” (current lines: 285-288).

Reviewer 2 Report

The paper by Bonente et al is very interesting and clinically relevant. It is performed on patients’ tissue samples of epiretinal membranes and interactomic analysis. The data are well presented and clear. The methodology is appropriate. The discussion is very well elaborated and conclusions are highlighted. I have only minor comments:

1.      Line  7…..Epiretinal membranes – please put (ERM) after Epiretinal membranes and to each section when first introduce abbreviation

2.      Line 9 ….x proteins. Recently, we have reviewed ERM – word recently is in bold this should be corrected

3.      The limitations of the study should be described in the Discussion section.

Author Response

We are submitting the revised version of the manuscript ijms-2411706 by Bonente et al. We are grateful to Reviewers for their comments as the suggested changes have improved the clarity of the manuscript.

A detailed point-by-point replay is reported below for each Reviewer #2 comment. All changes in the manuscript have been marked up by using the “Track Changes” function.

The paper by Bonente et al is very interesting and clinically relevant. It is performed on patients’ tissue samples of epiretinal membranes and interactomic analysis. The data are well presented and clear. The methodology is appropriate. The discussion is very well elaborated and conclusions are highlighted. I have only minor comments.

We are very grateful to Rev. #2 for her/his appreciation of our work.

  1. Line 7…..Epiretinal membranes – please put (ERM) after Epiretinal membranes and to each section when first introduce abbreviation.

Done, as suggested.

  1. Line 9 ….x proteins. Recently, we have reviewed ERM – word recently is in bold this should be corrected.

Done, as suggested.

  1. The limitations of the study should be described in the Discussion section.

We agree with the Reviewer and we added the following sentence in the discussion section: “Other experiments, e.g. immunoprecipitation, would be useful to prove direct interaction between endogenous PDPN and CD44. Nonetheless, the small amounts of proteins recovered from ERMs define consistent restrictions for direct assays on protein-protein interaction”, (current lines: 263-265). This sentence highlights as further experiments would be needed to prove a direct functional interaction between PDPN and CD44. Unfortunately, the few amounts of proteins that we can extract from ERMs consistently limits more targeted experiments. This is the main limit of our work.

Round 2

Reviewer 1 Report

Dear authors, most of the issues have been addressed properly. Still, I'm not a big fan of the in silico analysis. There is to much data included that itself is provided by in silico research again. It becomes more and more a kind of self-fulfilling system that misses the lab work. And whether that is even statistically relevant remains speculative. 

 none